# Towards Aligned Data Forgetting via Twin Machine Unlearning

## Abstract

Modern privacy regulations have spurred the evolution of machine unlearning, a technique enabling a trained model to efficiently forget specific training data. In prior unlearning methods, the concept of "data forgetting" is often interpreted and implemented as achieving *zero* classification accuracy on such data. Nevertheless, the authentic aim of machine unlearning is to achieve *alignment* between the unlearned model and the gold model, *i.e.*, encouraging them to have *identical* classification accuracy. On the other hand, the gold model often exhibits *non-zero* classification accuracy due to its generalization ability. To achieve aligned data forgetting, we propose a Twin Machine Unlearning (TMU) approach, where a twin unlearning problem is defined corresponding to the original unlearning problem. Consequently, the generalization-label predictor trained on the twin problem can be transferred to the original problem, facilitating *aligned data forgetting*. Additionally, we introduce a noise-perturbed fine-tuning scheme to balance the trade-off between retaining the model's generalization ability and enhancing its resilience to Membership Inference Attacks. Comprehensive empirical experiments illustrate that our approach significantly enhances the alignment between the unlearned model and the gold model. The code is available here.

## 1 Introduction

Machine learning model providers tend to collect extensive data from the Internet and utilize it to train their machine learning models. The recent introduction of data privacy and protection regulations (European Union's GDPR Regulation (2018), and California Consumer Privacy Act (CCPA) Goldman (2020)) obligate these model providers to comply with the request-to-delete Dang (2021) from the data owner. For example, a corporation offering facial recognition services might acquire facial images from the Internet to train their facial recognition models. Subsequently, a user might discover that the company has utilized his/her facial images in model training. In such a scenario, the user reserves the right to petition the company to revise the facial recognition model to forget his/her facial image data.

The straightforward solution is to entirely discard the trained model, delete his/her facial images from the training data, and re-train a new model from scratch (*i.e., called gold model*). Unfortunately, re-training from scratch is expensive. Therefore, machine unlearning Garg et al. (2020); Ginart et al. (2019); Cao & Yang (2015); Gupta et al. (2021); Sekhari et al. (2021); Brophy & Lowd (2021); Ullah et al. (2021) has garnered considerable attention, aiming to *efficiently revise* a trained model to forget a cohort of training data, without affecting the performance on the remaining data.

In most prior unlearning methods Graves et al. (2021); Tarun et al. (2023), the "forgetting" of a cohort data $D_f$ is commonly interpreted and implemented by decreasing the classification accuracy on $D_f$ *as much as possible*, *i.e.*, pursuing *zero* accuracy $ACC_{D_f}^{unlearn} = 0$. This implementation works well when forgetting an **entire** class $k$ (*i.e.*, $D_f = D_k$). However, it is not suitable when the aim is to forget only **a subset** of class $k$ (*i.e.*, $D_f \subset D_k$). In this case, even though $D_f$ are not involved in the training of the gold model, the remaining samples from class $k$ (*i.e.*, samples $\in (D_k - D_f)$) are still involved in training. Thus, the gold model has ability to recognize class $k$. Consequently, the gold model can still correctly classify a portion of $D_f$ due to its generalization ability, resulting in $ACC_{D_f}^{gold} \neq 0$. As we know, the actual aim of machine unlearning is to obtain an unlearned model that is well *aligned* with the gold model. Here, "align" implies **requiring the**

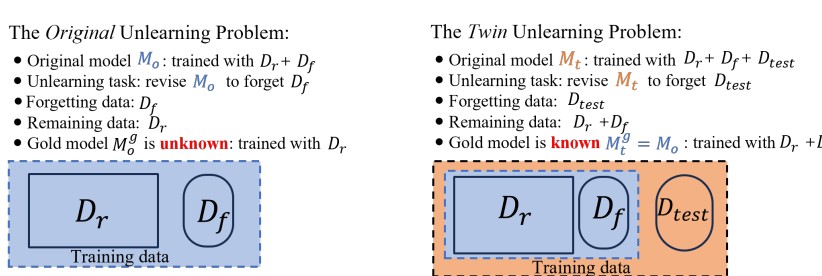

Figure 1: Construction of the *Twin Model* and corresponding *Twin Unlearning Problem*.

**unlearned model to possess the *identical* classification accuracy as the gold model**. Obviously, it is not appropriate to pursue $ACC_{D_f}^{unlearn} = 0$.

In this paper, we denote the samples $x \in D_f$ that can be correctly classified by the gold model as the '*easy*' samples. Conversely, the samples that cannot be correctly classified are referred to as the '*hard*' samples. As these easy/hard labels for $D_f$ are defined based on the model's generalization ability, we refer to them as *generalization-labels*. Therefore, **a well-aligned unlearning algorithm should specifically decrease the classification accuracy on *hard* samples rather than *all* samples in** $D_f$. In other words, it is essential to first identify the hard samples from $D_f$, and subsequently reduce the classification accuracy on such samples. However, since the gold model is **unknown**, obtaining these generalization-labels becomes challenging. This is the reason why prior unlearning methods have to adopt achieving *zero* classification accuracy as the objective of data forgetting.

In this paper, we aim to achieve aligned data forgetting. To this end, we propose to train a binary classifier to predict the generalization-labels for $D_f$. With these predicted labels, $D_f$ is partitioned into an easy subset $D_f^e$ and a hard subset $D_f^h$. Subsequently, we specifically decrease the classification accuracy on $D_f^h$, while retaining the classification accuracy on $D_f^e$. The most challenging aspect of our approach lies in training such a binary classifier on a specific dataset, which involves two sub-challenges: (1) Build a specific labeled dataset for training the binary classifier; (2) Construct a discriminative representation for the binary classifier.

To address the first challenge, we formulate a twin unlearning problem corresponding to the original one. Specifically, the original model $M_o$ is fine-tuned with $D_{test}$ to produce the ***Twin Model*** $M_t$. As a result, in the context of $M_t$, the original model $M_o$ can be seen as analogous to $M_t$'s gold model (*i.e.,* $M_t^g = M_o$), where $D_{test}$ can be seen as analogous to $M_t$'s forgetting data, as shown in Fig.1. In another word, given a trained model $M_t$, the ***Twin Unlearning Problem*** is to forget $D_{test}$ from $M_t$. Note that in the twin unlearning problem, the gold model is known $M_t^g = M_o$.

The reason for formulating such a twin unlearning problem is that we can easily obtain a labeled dataset – the dataset $D_{test}$. Since the gold model of the twin unlearning problem is known (*i.e., $M_o$*), we can easily obtain the generalization-labels for $D_{test}$, specifically by employing $M_o$ to make class predications on $D_{test}$.

Regarding the second challenge, we propose to integrate three complementary features, each possessing strong discriminability to distinguish between easy and hard samples. (1) The first feature is the Distance Feature (DF). For each sample $x$ in $D_f$, we identify its nearest neighbor in $D_r$, and the distance between them is considered as a feature. Intuitively, if $x$ is an easy sample, it typically has similar samples in $D_r$, resulting in a small distance. In contrast, the distance feature for a hard sample tends to be large. (2) The second feature is the Adversarial-attack Feature (AF). We construct this feature by employing adversarial attacksMadry et al. (2017); Goodfellow et al. (2014); Moosavi-Dezfooli et al. (2017), which aim to fool a trained model into making incorrect predictions with small adversarial perturbations. Clearly, hard samples are more vulnerable to adversarial attacks compared to easy samples due to their proximity to the decision boundary. Hence, the results of adversarial attack can be considered a discriminative feature.(3) The third feature is the Curriculum-learning-loss Feature (CF). Curriculum learning theory Kumar et al. (2010); Tullis & Benjamin (2011); Bengio et al. (2009) suggests that easy samples are learned earlier than hard samples. Therefore, the loss value of the first fine-tuning epoch is considered a feature to distinguish between easy and hard samples.

At last, in addition to reducing the classification accuracy on the hard subset $D_f^h$, it is equally crucial to maintain the model's generalization ability on the easy subset $D_f^e$, ensuring model can correctly classify those samples. A naive approach to achieving this is by directly minimizing their classification loss during machine unlearning. However, such an approach would leave the model vulnerable to Membership Inference Attack (MIA), which is commonly used to assess the success of data forgetting. To address this, we propose a noise-perturbed fine-tuning scheme. This scheme minimizes the classification loss on a *noise-perturbed* version of $D_f^e$. By doing so, we could strike a balance between preserving the model's generalization ability and enhancing its resilience to MIA.

Overall, we summarize our contributions as follows.

- We emphasize alignments in data forgetting and introduce a Twin Machine Unlearning (TMU) approach to enhance such alignments.

- We devise three discriminative features to distinguish easy and hard samples, employing adversarial attack and curriculum learning strategies.

- To withstand the assessment of data forgetting from MIA, we propose a noise-perturbed fine-tuning scheme to enhance model's resilience to MIA.

## 2 PROBLEM OF DATA FORGETTING

Let $D = \{x_i, y_i\}_{i=1}^N$ be a dataset of images $x_i$, each with a label $y_i \in \{1, \dots K\}$ representing a class. The original model $M_o(\theta)$ is trained with the $D$, which could be a DNN with parameters $\theta$.

Let $D_f$ be a subset of the $D$, whose information we want to forget from the trained model $M_o(\theta)$. The $D_f$ is called the *forgetting data*, which could be *all* or *some* of the data with a given label $k$, *i.e.,* corresponding to forgetting an entire class or a subset of a class, respectively. In this paper, we are particularly interested in the case of forgetting a subset of a class, as the alignment problem becomes more critical in this scenario. The *remaining data* is denoted by $D_r$, whose information is desired to be kept unchanged in the model. $D_f$ and $D_r$ together represent the entire training set $D$ and are mutually exclusive, *i.e.,* $D_r \cup D_f = D$ and $D_r \cap D_f = \emptyset$.

Since $D_f$ has been used to train the $M_o(\theta)$, the model parameters $\theta$ will contain information about $D_f$. The task of data forgetting aims to **forget the information of $D_f$ from the trained model** $M_o(\theta)$. The ideal solution is to train a new model from scratch with $D_r$, which is denoted as the *gold* model $M_o^g(\theta_r)$. However, obtaining the gold model is time-consuming due to the expensive re-training procedure. Thus, the practical data forgetting aims to efficiently revise the original model $\theta$ with a 'scrubbing/forgetting' function $s()$, so that the revised model $s(\theta)$ is as close to the gold model $\theta_r$ as possible. The $s()$ is often implemented by a *machine unlearning* algorithm.

### 2.1 ALIGNMENT IN DATA FORGETTING

The model alignment is often measured by some metrics (called *readout functions* Golatkar et al. (2021)), such as: (i) accuracy on the test set $D_{test}$, *i.e.,* $ACC_{D_{test}}$; (ii) accuracy on the forgetting data $D_f$, *i.e.,* $ACC_{D_f}$; (iii) accuracy on the remaining data $D_r$, *i.e.,* $ACC_{D_r}$. Achieving alignment on $ACC_{D_{test}}$ and $ACC_{D_r}$ is straightforward, as the target is to increase them as much as possible. In contrast, the challenge lies in aligning the $ACC_{D_f}$, *i.e.,* the unlearned model is desired to have the same classification accuracy as the gold model $ACC_{D_f}^{unlearn} = ACC_{D_f}^{gold}$. Since the gold model is unknown, obtaining its $ACC_{D_f}^{gold}$ is impossible, let alone achieving accuracy alignment.

As a result, prior unlearning methods tend to replace this objective with a surrogate objective, *i.e.,* decreasing the accuracy $ACC_{D_f}$ as much as possible. For the case of forgetting an entire class, this implementation is acceptable since, after removing an entire class, the gold model tends to have $ACC_{D_f}^{gold} = 0$, aligning with the surrogate objective. However, in the case of forgetting a subset of class, the remaining data will still contain samples of the forgetting class. Due to the generalization ability of gold model, the gold model may correctly classify some samples in $D_f$, *i.e.,* $ACC_{D_f}^{gold} \neq 0$. Thus, we have to carefully consider the alignment challenges.

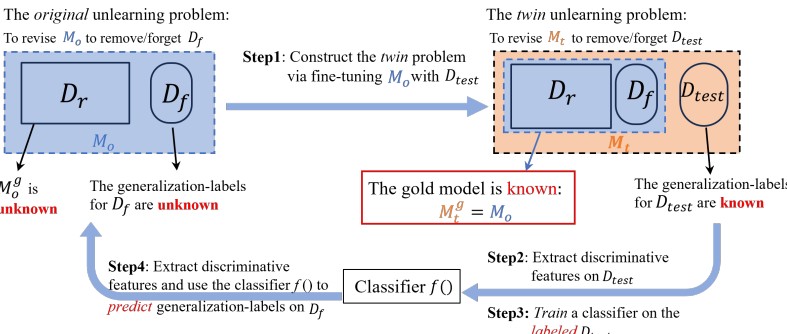

Figure 2: The workflow of our approach. We first construct the twin model by fine-tuning the $M_o$ with $D_{test}$ to produce $M_t$. And then, we extract discriminative features and train a binary classifier on $D_{test}$. Consequently, we transfer the binary classifier to the original problem to predict the generalization-labels on $D_f$. Finally, we reduce classification accuracy on $D_f^h$.

Furthermore, as highlighted by Golatkar et al. (2020a), implementing data forgetting by merely decreasing $ACC_{D_f}$ may give rise to other issues such as information exposure. For instance, it could lead to the *Streisand effect*. This effect describes unexpected model behavior on forget samples, potentially leaking information about that data. If the samples in $D_f$ are consistently misclassified, it might raise suspicious. Therefore, it is crucial to emphasize the alignment in data forgetting.

## 3 OUR APPROACH

### 3.1 TWIN UNLEARNING PROBLEM

In our approach, we first construct a new unlearning problem corresponding to the original one. As shown in the left part of Fig.2, for the original unlearning problem, we have an original model $M_o$ that is trained with $D_r+D_f$. Our task is to get an unlearned model $M_u$ from $M_o$, so that $M_u$ is well aligned with the gold model $M_o^g$. The $M_o^g$ is a model trained from scratch only with $D_r$.

As shown in the right part of Fig.2, if we fine-tune the $M_o$ with another dataset $D_{test}$, we obtain a new model, namely the *twin model* $M_t$. Consequently, we can construct a *twin unlearning problem*: we regard the $M_t$ as the "original model" for the twin problem, which is trained with $D_r+D_f+D_{test}$. We regard the $D_{test}$ as the "forgetting data". Thus, our twin unlearning problem is defined as:

> Given a trained model $M_t$, the unlearning task is to forget $D_{test}$ from $M_t$.

Obviously, the twin problem's "gold model" $M_t^g$ is known, which is exactly the original problem's original model $M_o$,

$$M_t^g = M_o \tag{1}$$

Regarding the constructed twin unlearning problem, since its gold model $M_t^g$ is known, we can easily obtain the ground-truth generalization-labels for its forgetting data $D_{test}$, *i.e.,* run the inference of the model $M_t^g$ over the $D_{test}$ to obtaining the classification results. So far, **we have build a specific labeled dataset.** We can train a binary classifier $f()$ to distinguish between hard and easy samples on $D_{test}$ for the twin problem.

It is usually to assume that $D_{test}$ and $D_f$ have independent and identical distribution (I.I.D). Thus, the binary classifier $f()$ can be transferred from the twin problem back to the original unlearning problem for predicting the generalization-labels on $D_f$. In this way, we can effectively partition $D_f$ into a hard subset $D_f^h$ and an easy subset $D_f^e$. Finally, when solving the original unlearning problem, we can fine-tune the original model $M_o$ by reducing classification accuracy on $D_f^h$ (*i.e.,* maximizing its loss value) while retaining the accuracy on $D_f^e$ (*i.e.,* maximizing its negative loss

value), as follows:

$$\max_{\theta}(L_{D_f^h}(x,y;\theta) - L_{D_f^e}(x,y;\theta)) \tag{2}$$

This naive unlearning approach can properly address the alignment challenge in data forgetting. However, our empirical study shows that it cannot afford the examination of Membership Inference Attack (MIA). MIA was initially introduced to determine whether a sample was included in a model's training set. More recently, it has been employed to assess the effectiveness of data forgetting. If the MIA asserts that a forgotten sample belongs to the training data, we conclude that the unlearning algorithm has failed to forget this data.

MIA typically operates by evaluating the loss value of a sample—if the loss is below a certain threshold, the sample is presumed to belong to the training set. In our naive unlearning approach with Eq(2), the loss for $D_f^e$ is minimized, which leads MIA to regard $D_f^e$ as part of the training set.

To withstand the examination of MIA, it is essential to convince MIA that $D_f^e$ has indeed been forgotten, while simultaneously ensuring that the unlearned model retains its ability to generalize and recognize $D_f^e$. To achieve this, we propose an enhanced unlearning approach as Eq.equation 3, designed to minimize the loss value for a noise-perturbed version of $D_f^e$, denoted as $\widetilde{D}_f^e$. This can be formulated as follows:

$$\max_{\theta}(L_{D_f^h}(x,y;\theta) - L_{\widetilde{D}_f^e}(\widetilde{x},y;\theta)) \tag{3}$$

$$\text{where } \widetilde{D}_f^e = \{\widetilde{x}|\widetilde{x} = x + r, x \in D_f^e\}$$

where a Gaussian noise $r$ is added to each sample in $D_f^e$.

Since the loss value for $D_f^e$ is not directly minimized, our enhanced approach can withstand the examination of MIA. At the same time, by minimizing the loss on $\widetilde{D}_f^e$—a noise-perturbed version of $D_f^e$—the unlearned model retains its generalization ability on $D_f^e$, as the introduced noise $r$ is not large. Particularly, there exists a balance between preserving the model's generalization ability and enhancing its resilience to MIA. We can make a trade-off by selecting an appropriate noise level $r$, increasing the noise level enhances resilience to MIA, while decreasing it improves the model's generalization ability.

## 3.2 DISCRIMINATIVE FEATURES

Besides building a specific labeled dataset for training the binary classifier, another core of our approach is **to construct discriminative features for the classifier**. Briefly, we propose to integrate three complementary features by employing adversarial attack and curriculum learning strategies.

### 3.2.1 DISTANCE FEATURE (DF)

In the case of forgetting a subset of a class $k$, the remaining data $D_r$ will still contain samples belonging to the forgetting class $k$. Therefore, for each sample $x \in D_f$, we can find some samples $x' \in D_r$ similar to $x$. The similarity can be measured by the distance $l(x, x')$ with respect to a feature extractor $g()$,

$$l(x, x') = |g(x) - g(x')| \tag{4}$$

For each $x \in D_f$, we can identify the top-$N$ similar samples $x_i' \in D_r, (i = 0, \ldots, N - 1)$. Intuitively, these distances $l(x, x_i')$ for an easy sample would be smaller than that for a hard sample. Thus, the concatenation of these top-$N$ distances $l(x, x_i'), (i = 0, \ldots, N - 1)$ can be regarded as a discriminative feature to distinguish between easy and hard samples. In practice, we adopt the activation value of the penultimate layer (*i.e.,* the layer just before the last fully-connected layer) as the $g()$.

### 3.2.2 ADVERSARIAL-ATTACKING FEATURE (AF)

Whether a sample can be correctly generalized or not (*i.e.,* belong to a easy or hard sample) is closely related to the positional relation between it and the classification boundary. If it is near the boundary,

it is hard to correctly classify it with high confidence. Thus, the easy samples often stay far from the boundary, while the hard samples tend to stay near the boundary.

In order to leverage such positional relation knowledge to distinguish between easy and hard samples, we employ the untargeted adversarial attack technique. As we know, given any sample $x$, we can generate a corresponding adversarial sample $\tilde{x}$ whose classification result is different from its ground-truth label. The adversarial sample $\tilde{x}$ is often generated by perturbing $x$ with a certain perturbation $r$. The perturbation $r$ is typically constrained by a certain attack budget $\epsilon$, as follows,

$$\max_{r} L(\widetilde{x}, y; \theta) \tag{5}$$
$$\text{s.t. } ||r||_p < \epsilon, \widetilde{x} = x + r$$

The essence of adversarial attack is to move $x$ across the classification boundary, *i.e.,* changing the classification result. Thus, given a sufficiently large budget $\epsilon$, we can successfully conduct the adversarial attack on any sample, *i.e.,* successfully generating $\tilde{x}$.

Intuitively, the attack budget for a sample is related to the positional relation between it and the classification boundary. If the sample is near the boundary, a small budget is enough to conduct a successful attack. Thus, we can distinguish between hard and easy samples like that: we use *a relatively small attack budget* to conduct adversarial attack, and we identify easy and hard samples based on whether the attack is successful or fails, *i.e.,* hard samples can be successfully attacked with a small attack budget, since it is near the classification boundary.

Furthermore, the bipolar results about successful or failure attack cannot be regarded as a good continuous discriminative feature. Instead, we compute the classification score/logits $s(\tilde{x})$ for the adversarial sample $\tilde{x}$. On the other hand, we calculate the classification score $s(x)$ for the clean sample $x$, and measure the cross-entropy between $s(\tilde{x})$ and $s(x)$,

$$dist(x, \tilde{x}) = H(s(\tilde{x}), s(x)) \tag{6}$$

Obviously, the $dist(x, \tilde{x})$ will be large if the attack is successful and vice versa. The larger the $dist(x, \tilde{x})$ is, the higher the likelihood of a successful attack. Thus, the $dist(x, \tilde{x})$ is regarded as the second discriminative feature to distinguish between easy and hard samples.

### 3.2.3 CURRICULUM-LEARNING-LOSS FEATURE (CF)

Curriculum learning theory suggests that easy samples are learned earlier than hard samples. Thus, we propose to employ a curriculum learning strategy to build our third feature. Curriculum learning has illustrated that neural networks tend to learn easy samples very quickly at the beginning of the training iterations. Conversely, the hard samples are learned at the later iterations. Specifically, we train a *randomly-initialized* network $M_r$ from scratch, where the architecture of $M_r$ is the same as the architecture of original model $M_o$. To distinguish between easy and hard samples, we just train the model $M_r$ for *one* or *two* epochs, instead of completing the full training procedure.

Regarding whether a sample $x$ has been well-learned by $M_r$, we use the loss $loss(x)$ as a metric, *i.e.,* a small loss value implies that $x$ has been well learned (indicating it is a easy sample) while a large loss value implies that $x$ has not been well learned (indicating it is a hard sample). Therefore, the curriculum-learning-loss $loss(x)$ is considered as the third feature to distinguish between easy and hard samples.

Note that the model $M_r$ is trained with a part of $D_r + D_f$ and $D_{test}$. In practice, we find that it is enough to train $M_r$ by just utilizing 30% of all $D_r + D_f$. Thus, the training of $M_r$ is much cheaper than the training of $M_o$.

### 3.3 BINARY CLASSIFIER

We have build a specific labeled dataset $D_{test}$. On the other hand, we have devised three discriminative features to distinguish between easy and hard samples. Next, we will train a binary classifier to distinguish between easy and hard samples with these feature over the labeled dataset. Specifically, we employ an MLP network with two hidden layers as the binary classifier. We concatenate the three discriminative features as the input of the binary classifier. Note that we have opted for a straightforward binary classifier due to the discriminative nature of our features and our desire to keep the unlearning process cost-effective.

## 4 EVALUATION

### 4.1 EXPERIMENTAL SETTING

**Datasets & Models.** We evaluate our approach using three public image datasets: CIFAR-10, CIFAR-100 Krizhevsky et al. (2009) and VGGFaces2 Cao et al. (2018). CIFAR-10 comprises 10 classes, and we perform data forgetting evaluations for each class independently. For CIFAR-100, we randomly select 17 out of 100 classes for evaluation due to space constraints. Since we emphasize on forgetting only a subset of one class, we randomly select 100 images as $D_f$. For VGGFaces2, we randomly select 10 out of 100 celebrities as the forgetting class, and then randomly choose 50 facial images to comprise $D_f$ for each forgetting class.

Three common image classification neural networks are employed for evaluation: ResNet-18 He et al. (2016), AllCNN Springenberg et al. (2014), and Vision Transformer Lee et al. (2021). The original models are trained for 200 epochs using Stochastic Gradient Descent (SGD) optimizer with a momentum of 0.9, weight decay of $5e-4$, and an initial learning rate of 0.01. The learning rate is divided by 10 after 100 and 150 epochs.

**Baseline Methods.** We compare our approach against four machine unlearning methods: (1) *Negative Gradient* Golatkar et al. (2020a): fine-tune the original model on $D$ by increasing the loss for samples in $D_f$, which is the common surrogate objective adopted by most unlearning methods. (2) *Fine-tuning*: fine-tune the model on $D_r$ using a slightly large learning rate. This is analogous to catastrophic forgetting, as fine-tuning without $D_f$ may cause the model to forget $D_f$. (3) *Random Labeling* Graves et al. (2021): fine-tune the model on $D$ by assigning random labels to samples in $D_f$, causing those samples to receive a random gradient. (4) *Bad Teacher* Chundawat et al. (2023): it explores the utility of competent and incompetent teachers in a student-teacher framework to induce forgetfulness. Note that this work is closely related to our approach since it also emphasizes alignments in data forgetting.

**Evaluation Metrics.** We assess the alignment quality of unlearning methods using three metrics: (1) *Accuracy on $D_f$ and $D_{test}$*: Ideally, the unlearned model is expected to achieve the same accuracy as the gold model. Let $ACC_{D_f}$ and $ACC_{D_f}^g$ denote the accuracy of the unlearned model and the gold model on $D_f$. Thus, the differences between them, $\Delta = |ACC_{D_f} - ACC_{D_f}^g|$, can be considered as the measure of alignment quality. Additionally, accuracy on $D_{test}$ is employed to assess whether the model accuracy is compromised after data forgetting. (2) *Activation Distance*: This metric calculates the average L2-distance between the activation values of the unlearned model and the gold model on $D_f$. A smaller activation distance indicates a better alignment between two models. (3) *Membership Inference Attack (MIA)*: MIA aims to determine whether a given sample was included in a model's training set. Consequently, if we have an ideal MIA, we can employ it to evaluate the success of data forgetting. The lower the success rate of the MIA, the more effective the data forgetting process. Thus, the attack success rate (ASR) is employed as a key evaluation metric for data forgetting effectiveness. Although a perfect MIA is unattainable, we utilize a state-of-the-art MIA approach Ye et al. (2022) in our evaluation.

#### 4.1.1 IMPLEMENTATION DETAILS

All our experiments were conducted on a single NVIDIA RTX 3090 GPU. For the binary classifier, we employ an MLP network with two hidden layers, where the sizes of the hidden layers are 64 and 32, respectively. We use the ReLU activation function in the classifier. The classifier is trained for 100 epochs using SGD with fixed learning rate of 0.01, momentum 0.9 and weight decay 0.0005. For the Adversarial-attacking Feature (AF), we set the attack budget as $\epsilon = 4/255$. For the Curriculum-learning-loss Feature (CF), we obtain it from the loss on the models trained from scratch for two epochs using $D_f$ and $D_{test}$, as well as a 30 % random subset of $d_r$. The learning rate is set to one-tenth of the learning rate used during the fine-tuning process. The loss feature is obtained from the same model used for $D_f$.

### 4.2 MAIN RESULTS

We compare our approach with four state-of-the-art unlearning methods. Table.1 presents the comparison on the CIFAR-10 dataset. When adopting the ResNet-18 neural network, the original model

Table 1: Comparison with four unlearning methods on CIFAR-10. Three metrics are adopted: (1) Accuracy on $D_{test}$ and $D_f$; (2) Activation Distance; (3) MIA. We conduct the data forgetting experiment for each class in CIFAR-10 independently, where 100 images are randomly selected as $D_f$. The results in this table are the **average values** over all 10 classes in CIFAR-10.

| Experiment | | Accuracy on $D_{test}$ and $D_f$ | | | Activation Distance | Member Inference Attack |
|---|---|---|---|---|---|---|
| Model | Methods | $ACC_{D_{test}}\uparrow$ | $ACC_{D_f}$ | $\Delta\downarrow$ | $AD\downarrow$ | $ASR\downarrow$ |
| ResNet-18 | Re-training (Gold Model) | 85.31 | 86.9 | 0 | 0 | 39.1 |
| | Fine-tuning | 85.06 | 100 | 13.1 | 0.58 | 53.2 |
| | Negtive Gradient | 84.81 | 10.6 | 76.3 | 0.72 | 7.3 |
| | Random Label | 85.43 | 0 | 86.9 | 1.25 | 0 |
| | Bad Teacher | 82.37 | 91.5 | 8.0 | 0.71 | 23.5 |
| | Ours | 84.86 | 90.8 | 4.2 | 0.49 | 31.6 |
| ALLCNN | Re-training (Gold Model) | 86.58 | 86.4 | 0 | 0 | 55.9 |
| | Fine-tuning | 87.34 | 100.0 | 13.1 | 0.32 | 45.6 |
| | Negtive Gradient | 86.87 | 3.3 | 84.3 | 0.84 | 1.7 |
| | Random Label | 86.89 | 0 | 86.4 | 0.96 | 0 |
| | Bad Teacher | 85.96 | 88.0 | 5.8 | 0.49 | 56.4 |
| | Ours | 86.78 | 87.7 | 4.9 | 0.29 | 26.6 |
| Vit | Re-training (Gold Model) | 84.61 | 84.6 | 0 | 0 | 32.1 |
| | Fine-tuning | 84.83 | 100.0 | 15.4 | 0.65 | 80.0 |
| | Negtive Gradient | 84.51 | 62.7 | 23.9 | 0.92 | 32.5 |
| | Random Label | 84.21 | 2.4 | 82.2 | 0.88 | 1.6 |
| | Bad Teacher | 83.05 | 87.9 | 5.3 | 0.59 | 50.2 |
| | Ours | 84.07 | 86.1 | 3.3 | 0.40 | 51.5 |

Table 2: Comparison with four unlearning methods on CIFAR-100.

| Experiment | | Accuracy on $D_{test}$ and $D_f$ | | | Activation Distance | Member Inference Attack |
|---|---|---|---|---|---|---|
| Dataset | Methods | $ACC_{D_{test}}\uparrow$ | $ACC_{D_f}$ | $\Delta\downarrow$ | $AD\downarrow$ | $ASR\downarrow$ |
| Resnet-18 | Re-training (Gold Model) | 81.03 | 81.29 | 0 | 0 | 65.66 |
| | Fine-tuning | 80.61 | 100 | 18.71 | 0.56 | 79.72 |
| | Negtive Gradient | 80.27 | 23.94 | 57.37 | 0.72 | 10.9 |
| | Random Label | 80.09 | 14.35 | 66.94 | 0.83 | 8.2 |
| | Bad Teacher | 78.45 | 60.64 | 19.35 | 0.41 | 64.35 |
| | Ours | 79.55 | 89.29 | 7.94 | 0.28 | 37.83 |
| Vit | Re-training (Gold Model) | 82.00 | 84.71 | 0 | 0 | 45.94 |
| | Fine-tuning | 83.53 | 100.0 | 15.29 | 0.50 | 75.0 |
| | Negtive Gradient | 80.50 | 39.20 | 45.51 | 0.75 | 16.3 |
| | Random Label | 83.10 | 0 | 84.71 | 0.74 | 0 |
| | Bad Teacher | 81.19 | 81.47 | 8.64 | 0.25 | 75.27 |
| | Ours | 82.00 | 88.76 | 6.23 | 0.19 | 62.16 |

attains an average accuracy of $85.37\%$ across 10 classes. When adopting the AllCNN neural network, the original model has an average accuracy of $86.56\%$. In this evaluation, we conduct the data forgetting experiment for each class independently. For each class, 100 samples are randomly selected as $D_f$. Due to space constraints, the class-wise results are shown in Appendixes, while Table.1 illustrates the average results across 10 classes.

**Accuracy on $D_f$ and $D_{test}$:** We provide the accuracy of the gold model on both $D_{test}$ and $D_f$ (*i.e.,* $ACC_{D_{test}}^g$ and $ACC_{D_f}^g$). For each method, we provide the $ACC_{D_{test}}$, $ACC_{D_f}$, and $\Delta = |ACC_{D_f} - ACC_{D_f}^g|$. The closer the accuracy between the gold model and the unlearned model (*i.e.,* the smaller the $\Delta$ is), the better the alignment is achieved. In addition, the accuracy on $D_{test}$ is employed to assess whether the model's normal performance is compromised after data forgetting. The higher the accuracy on $D_{test}$ is, the better the normal performance is maintained.

From Table.1, it's evident that the Fine-tuning method cannot effectively accomplish data forgetting, as the $ACC_{D_f}$ remains almost at $100\%$. It implies that Fine-tuning can only forget an entire class but struggles to forget a subset of a class. In contrast, the Random Labeling method tends to misclassify all samples in $D_f$, resulting in $ACC_{D_f} = 0\%$. The Negative Gradient method performs better than Fine-tuning and Random Labeling, with an average difference of $\Delta_{NG} = 76.3\%$. Due to explicitly addressing the alignment issue, the Bad Teacher method makes significant progress, achieving a

Table 3: Ablation study on three discriminative features. We use the accuracy of predicted generalization-labels $ACC_{gl}$ as the metric to measure the quality of alignment.

| Forgetting Class | DF+AF+CF (Ours) | | DF | | AF | | CF | |
|---|---|---|---|---|---|---|---|---|
| | $ACC_{D_{test}}\uparrow$ | $ACC_{gl}\uparrow$ | $ACC_{D_{test}}\uparrow$ | $ACC_{gl}\uparrow$ | $ACC_{D_{test}}\uparrow$ | $ACC_{gl}\uparrow$ | $ACC_{D_{test}}\uparrow$ | $ACC_{gl}\uparrow$ |
| Class 0 | 90.4 | **92** | 89.2 | 92 | 90.1 | 92 | 89.2 | 91 |
| Class 1 | 94.4 | **100** | 93.8 | 92 | 93.7 | 97 | 94.3 | 100 |
| Class 2 | 79.6 | **84** | 72.6 | 83 | 71.1 | 85 | 75.1 | 82 |
| Class 3 | 76.9 | 79 | 69.2 | 72 | 65.5 | 69 | 76.6 | **81** |
| Class 4 | 83.3 | **85** | 80.3 | 80 | 82.3 | 82 | 83.7 | 82 |
| Class 5 | 82.9 | 82 | 78.5 | 77 | 77.0 | 74 | 81.5 | **86** |
| Class 6 | 91.7 | 89 | 88.2 | 85 | 89.3 | 82 | 91.0 | **90** |
| Class 7 | 90.8 | **87** | 91.1 | 86 | 91.7 | 86 | 91.7 | 84 |
| Class 8 | 94.7 | **92** | 92.9 | 90 | 82.7 | 89 | 94.7 | 91 |
| Class 9 | 93.6 | **98** | 92.0 | 94 | 92.0 | 95 | 93.2 | 96 |

remarkable average difference of $\Delta_{BadT} = 8\%$. Nevertheless, our approach outperforms the Bad Teacher method in terms of both $\Delta$ and $ACC_{D_{test}}$. Particularly, we improve the average difference $\Delta$ from 8% to 4.2%.

The comparison on the CIFAR-100 is shown in Table.2. We draw similar conclusions as with CIFAR-10. The Random Labeling method no longer misclassifies all samples in $D_f$, with an average $ACC_{D_f} = 14.35\%$. Our approach demonstrates a greater advantage than the Bad Teacher method, with $\Delta_{ours} = 7.94\%$ against $\Delta_{BadT} = 19.35\%$.

The comparison on the VGGFaces2 dataset is presented in Appendixes. Similarly, our approach consistently outperforms all other methods.

**Activation Distance:** Activation distance Golatkar et al. (2021) is another effective metric for evaluating the alignment in unlearning. A smaller activation distance indicates a higher similarity between the unlearned model and the gold model. From Table.1, our approach exhibits good performance, outperforming other methods.

**Membership Inference Attack (MIA):** We leverage MIA to evaluate the success of data forgetting. The lower the MIA's Attack Success Rate (ASR), the more successful the data forgetting process. In practice, we utilize an enhanced Membership Inference Attack Ye et al. (2022) in our evaluation. From Table.1, we can see that our approach can afford the examination of data forgetting from MIA. It is worthy to note that the Negative Gradient achieve better performance than our approach in terms of MIA assessment. This is because the Bad Teacher inherently aims to disrupt accuracy on $D_f$, which naturally leads to the resilience to MIA.

In our experiments, three different neural network architectures are evaluated, including ResNet-18 He et al. (2016), AllCNN Springenberg et al. (2014) and ViT Lee et al. (2021). From Table.1 and 2, we found that the performance of unlearning methods is less influenced by the neural network architecture. Specifically, the Fine-tuning method maintains $ACC_{D_f} = 100\%$, whereas in contrast, the Random Labeling method results in almost $ACC_{D_f} = 0\%$. Both methods fail to achieve effective data forgetting, regardless of the employed network architecture. We did not present the results for AllCNN on CIFAR-100, as AllCNN is unable to effectively handle a 100-class classification problem.

The Negative Gradient method exhibits relatively better performance, achieving an average difference $\Delta_{NG}^{AllCNN} = 84.3\%$ on AllCNN and $\Delta_{NG}^{ViT} = 23.9\%$ on ViT, respectively. In contrast, the Bad Teacher method makes more progress, achieving a remarkable average difference $\Delta_{BadT}^{AllCNN} = 5.8\%$ on AllCNN and $\Delta_{BadT}^{ViT} = 5.3\%$ on ViT, respectively. Nevertheless, our method still outperforms the Bad Teacher. Particularly, we improve the average difference $\Delta$ from 5.8% to 4.9% on AllCNN and $\Delta$ from 5.3% to 3.3% on ViT, respectively.

### 4.3 ABLATION STUDY

In our approach, three discriminative features are proposed and combined to distinguish hard samples from easy samples. We will evaluate their individual contributions through an ablation study in this section. Since these features are developed to predict the generalization-labels on $D_f$, we use the prediction accuracy as the metric. Note that the prediction accuracy $ACC_{gl}$ is positively related

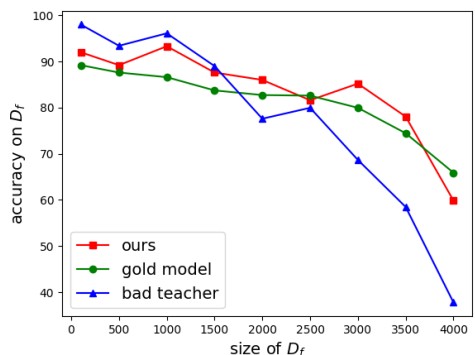

Figure 3: The quality of alignment varies with the increase in the size of $D_f$.

Table 4: Trade-off between preserving model's generalization and enhancing its resilience to MIA

| Noise Level | 0 | 1 | 3 | 5 | 10 |
|---|---|---|---|---|---|
| ACC | 87.2 | 85.14 | 84.05 | 79.18 | 76.46 |
| MIA | 71.0 | 55 | 37 | 33 | 27 |

to the quality of alignment (*i.e.,* $\Delta$, AD, and ASR). The higher the accuracy, the better the alignment. Table.3 shows the experimental results on CIFAR-10, where the original model adopts ResNet-18. From Table.3, the Curriculum-learning-loss Feature (CF) is slightly more discriminative than the Distance Feature (DF) and Adversarial-attacking Feature (AF). Nonetheless, the three features are complementary to each other.

We propose a noise-perturbed fine-tuning scheme to enhance model's resilience to MIA. In another ablation study, we compare the naive baseline approach (as the Eq(2)) with our approach (as the Eq(3)). From Table 4, we can see that our propose scheme enables the unlearned model to better withstand the assessment of data forgetting from MIA. Moreover, we can make a trade-off between preserving the model's generalization ability and enhancing its resilience to MIA by adjusting the noise level $r$. From Table 4, it turns out that increasing the noise level enhances resilience to MIA (measured with $ASR$), while decreasing it improves the model's generalization ability (measured with $ACC_{D_f}$).

### 4.4 FORGETTING MORE DATA

In previous experiments, we evaluate our approach for the situation that the size of forgetting data is small, *i.e.,* 100 images. In this section, we increase the size of $D_f$ from 100 to 4000 gradually, aiming to evaluate our approach under the situation of forgetting a larger amount of data.

We assess the performance by gradually increasing the size of $D_f$ to 500, 1000, 2000, 3000 and 4000. The performance is measured by the accuracy of predicted generalization-labels. From Fig.3, we can see that the alignment becomes more challenging with the increase of $D_f$ size. However, our approach consistently outperforms the Bad Teacher method, and its advantages become more pronounced with the increase in the size of $D_f$.

### 5 CONCLUSION

This paper aims at the task of data forgetting, emphasizing the alignment in data forgetting. We introduce a Twin Machine Unlearning approach, where a twin unlearning problem is constructed and leveraged to solve the original problem. Furthermore, three discriminative features are devised by employing adversarial attack and curriculum learning strategies. Additionally, to withstand the assessment of data forgetting from Membership Inference Attack, we propose a noise-perturbed fine-tuning scheme to enhance model's resilience to MIA. Extensive empirical experiments show that our approach significantly improve the alignment in data forgetting.

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

# A    APPENDIX

## A.1    CLASS-WISE RESULTS FOR DATA FORGETTING

### A.1.1    RESULTS ON CIFAR-10 AND CIFAR-100 DATASET

We conduct the data forgetting experiment for each class in CIFAR-10 and CIFAR-100 independently, as depicted in each row of Table.5, Table.6, Table.7, Table.8 and Table.9. For each class, 100 samples are randomly selected as $D_f$.

Table 5: Comparison with **ResNet-18** on **CIFAR-10** dataset. We conduct the data forgetting experiment for each class in CIFAR-10 independently, where 100 images are randomly selected as $D_f$.

| Removal Class | Gold model $ACC_{D_{test}}$ | Gold model $ACC_{D_f}$ | Fine-tuning $ACC_{D_{test}}$ | Fine-tuning $ACC_{D_f}$ | Negative Gradient $ACC_{D_{test}}$ | Negative Gradient $ACC_{D_f}; \Delta$ | Random Labeling $ACC_{D_{test}}$ | Random Labeling $ACC_{D_f}$ | Bad Teacher $ACC_{D_{test}}$ | Bad Teacher $ACC_{D_f}; \Delta$ | Ours $ACC_{D_{test}}$ | Ours $ACC_{D_f}; \Delta$ |
|---|---|---|---|---|---|---|---|---|---|---|---|---|
| Class 0 | 85.61 | 92 | 85.08 | 100 | 85.29 | 8; 84 | 85.49 | 0 | 83.98 | 95; 3 | 84.60 | 96; 4 |
| Class 1 | 85.27 | 97 | 85.07 | 100 | 85.31 | 11; 86 | 85.75 | 0 | 80.45 | 91; 6 | 84.76 | 98; 1 |
| Class 2 | 85.66 | 84 | 84.93 | 100 | 85.03 | 11; 73 | 85.03 | 0 | 84.27 | 73; 11 | 84.65 | 85; 1 |
| Class 3 | 84.8 | 74 | 84.97 | 100 | 85.09 | 9; 65 | 85.58 | 0 | 80.51 | 80; 6 | 85.07 | 72; 2 |
| Class 4 | 85.24 | 82 | 84.94 | 100 | 85.29 | 15; 67 | 85.22 | 0 | 81.96 | 94; 12 | 84.86 | 92; 10 |
| Class 5 | 85.49 | 76 | 85.03 | 100 | 85.06 | 13; 63 | 85.37 | 0 | 82.69 | 98; 22 | 84.43 | 81; 5 |
| Class 6 | 85.43 | 88 | 84.98 | 100 | 85.31 | 9; 79 | 85.62 | 0 | 82.64 | 98; 10 | 85.02 | 85; 3 |
| Class 7 | 84.97 | 87 | 84.90 | 100 | 85.18 | 9; 78 | 85.63 | 0 | 82.87 | 92; 5 | 85.10 | 99; 12 |
| Class 8 | 85.13 | 94 | 85.02 | 100 | 81.34 | 2; 92 | 85.42 | 0 | 83.11 | 97; 3 | 85.04 | 96; 2 |
| Class 9 | 85.53 | 95 | 85.12 | 100 | 85.25 | 19; 76 | 85.32 | 0 | 82.93 | 97; 2 | 84.49 | 97; 2 |
| Avg | 85.31 | 86.9 | 85.06 | 100 | 84.81 | 10.6; **76.3** | 85.43 | 0 | 82.37 | 91.5; **8** | 84.86 | 90.1; **4.2** |

Table 6: Comparison with **ResNet-18** on **CIFAR-100** dataset. We randomly sample 17 classes for evaluation, where 100 images are randomly selected as $D_f$.

| Removal Class | Gold model | | Fine-tuning | | Negative Gradient | | Random Labeling | | Bad Teacher | | Ours | |
|---|---|---|---|---|---|---|---|---|---|---|---|---|
| | $ACC_{D_{test}}$ | $ACC_{D_f}$ | $ACC_{D_{test}}$ | $ACC_{D_f}$ | $ACC_{D_{test}}$ | $ACC_{D_f};\Delta$ | $ACC_{D_{test}}$ | $ACC_{D_f};\Delta$ | $ACC_{D_{test}}$ | $ACC_{D_f};\Delta$ | $ACC_{D_{test}}$ | $ACC_{D_f};\Delta$ |
| road | 81.07 | 92 | 80.61 | 100 | 80.62 | 30; 62 | 80.00 | 52; 89 | 78.11 | 91; 1 | 78.97 | 99; 7 |
| turtle | 81.04 | 79 | 80.65 | 100 | 80.44 | 25; 54 | 80.24 | 7; 72 | 78.91 | 25; 54 | 79.09 | 84; 5 |
| chimpanzee | 81.48 | 90 | 80.64 | 100 | 80.82 | 28; 62 | 80.11 | 8; 82 | 77.55 | 70; 20 | 79.69 | 96; 6 |
| orchid | 81.15 | 86 | 80.60 | 100 | 80.48 | 30; 56 | 80.17 | 17; 69 | 78.64 | 80; 6 | 79.68 | 99; 13 |
| rabbit | 81.25 | 70 | 80.54 | 100 | 80.31 | 17; 53 | 79.99 | 1; 69 | 78.15 | 23; 47 | 79.17 | 91; 21 |
| forest | 81.33 | 72 | 80.64 | 100 | 80.31 | 22; 50 | 79.80 | 17; 55 | 79.24 | 74; 2 | 79.71 | 89; 17 |
| possum | 80.40 | 76 | 80.57 | 100 | 79.82 | 1; 75 | 80.02 | 11; 65 | 77.96 | 59; 17 | 79.84 | 84; 8 |
| fox | 81.18 | 89 | 80.65 | 100 | 80.38 | 23; 66 | 79.94 | 14; 75 | 78.43 | 64; 25 | 80.21 | 91; 2 |
| house | 81.76 | 85 | 80.47 | 100 | 80.67 | 24; 61 | 79.79 | 9; 76 | 78.11 | 74; 11 | 79.97 | 85; 0 |
| mushroom | 81.04 | 79 | 80.67 | 100 | 80.53 | 21; 58 | 79.98 | 10; 69 | 78.99 | 64; 15 | 79.4 | 96; 17 |
| chair | 81.05 | 90 | 80.50 | 100 | 80.74 | 39; 51 | 79.96 | 34; 56 | 78.37 | 85; 5 | 79.75 | 94; 4 |
| tiger | 80.79 | 86 | 80.74 | 100 | 80.51 | 30; 56 | 80.21 | 6; 80 | 78.58 | 59; 27 | 78.45 | 96; 10 |
| snail | 81.29 | 82 | 80.56 | 100 | 70.10 | 15; 67 | 80.21 | 4; 78 | 78.21 | 51; 31 | 79.77 | 84; 2 |
| worm | 80.92 | 90 | 80.72 | 100 | 80.34 | 41; 49 | 80.28 | 26; 64 | 78.56 | 87; 3 | 80.10 | 90; 0 |
| beetle | 81.16 | 82 | 80.61 | 100 | 80.66 | 27; 55 | 80.06 | 17; 65 | 78.74 | 64; 18 | 79.42 | 84; 2 |
| beaver | 81.21 | 54 | 80.67 | 100 | 80.32 | 6; 48 | 80.31 | 5; 49 | 78.33 | 27; 27 | 79.36 | 74; 20 |
| bed | 79.36 | 80 | 80.54 | 100 | 80.66 | 28; 52 | 80.41 | 6; 68 | 78.8 | 60; 20 | 79.85 | 81; 1 |
| Avg | 81.03 | 81.29 | 80.61 | 100 | 80.27 | 23.94; **57.37** | 80.09 | 14.35; **66.94** | 78.45 | 60.64; **19.35** | 79.55 | 89.29; **7.94** |

Table 7: Comparison with **AllCNN** on **CIFAR-10** dataset. Regarding $M_o$, its average accuracy over 10 classes is $87.21\%$.

| Removal Class | Gold model | | Fine-tuning | | Negative Gradient | | Random Labeling | | Bad Teacher | | Ours | |
|---|---|---|---|---|---|---|---|---|---|---|---|---|
| | $ACC^g_{D_{test}}$ | $ACC^g_{D_f}$ | $ACC_{D_{test}}$ | $ACC_{D_f}$ | $ACC_{D_{test}}$ | $ACC_{D_f};\Delta$ | $ACC_{D_{test}}$ | $ACC_{D_f}$ | $ACC_{D_{test}}$ | $ACC_{D_f};\Delta$ | $ACC_{D_{test}}$ | $ACC_{D_f};\Delta$ |
| Class 0 | 87.0 | 94 | 87.04 | 100 | 86.69 | 3; 91 | 86.96 | 0 | 86.09 | 86; 8 | 87.03 | 89; 5 |
| Class 1 | 86.58 | 97 | 87.49 | 100 | 86.86 | 2; 95 | 86.99 | 0 | 86.32 | 92; 5 | 87.43 | 98; 1 |
| Class 2 | 87.09 | 82 | 87.42 | 100 | 86.68 | 5; 77 | 86.53 | 0 | 85.71 | 75; 7 | 87.06 | 87; 5 |
| Class 3 | 86.22 | 78 | 87.38 | 100 | 86.85 | 2; 76 | 86.62 | 0 | 86.54 | 80; 2 | 87.51 | 69; 9 |
| Class 4 | 86.4 | 87 | 87.39 | 100 | 86.89 | 3; 84 | 87.72 | 0 | 84.45 | 96; 9 | 86.46 | 92; 5 |
| Class 5 | 86.36 | 79 | 87.23 | 100 | 86.75 | 5; 74 | 86.52 | 0 | 85.98 | 86; 7 | 87.37 | 83; 4 |
| Class 6 | 86.46 | 87 | 87.42 | 100 | 87.00 | 1; 86 | 87.39 | 0 | 86.44 | 91; 4 | 86.86 | 89; 2 |
| Class 7 | 86.24 | 82 | 87.31 | 100 | 86.79 | 1; 81 | 87.24 | 0 | 86.53 | 91; 9 | 87.24 | 89; 7 |
| Class 8 | 87.0 | 91 | 87.65 | 100 | 87.22 | 6; 85 | 86.95 | 0 | 84.51 | 90; 1 | 86.84 | 87; 4 |
| Class 9 | 86.54 | 87 | 87.16 | 100 | 86.93 | 5; 82 | 86.02 | 0 | 87.05 | 93; 6 | 87.05 | 94; 7 |
| Avg | 86.58 | 86.4 | 87.34 | 100 | 86.87 | 3.3; **83.1** | 86.89 | 0 | 85.96 | 88.0; **5.8** | 86.78 | 87.7; **4.9** |

### A.1.2 RESULTS ON VGGFACES2 DATASET

We further apply our method to a real-world facial dataset, VGGFace2. We randomly select 10 classes of faces from the VGGFace2 dataset, randomly sampled with at least 500 images each. The dataset is divided into train and test sets in a 7:3 ratio. For each class, we conduct a data removal experiment where 100 samples are randomly selected as $D_f$. The performance of our method is depicted in each row of Table.10. We observe that our method still outperforms other methods, achieving the minimum $\bar{\Delta}$ in each class, with the average $\bar{\Delta} = 3\%$.

### A.2 FORGETTING MORE DATA

To assess the performance when forgetting a larger amount of data, we evaluate our method with the size of $D_f$ as 500 and 4000, respectively. From Table.11 and Table.12, despite a slight decrease in performance compared to forgetting 100 images, our method still outperforms others. Our method outperforms the *Bad Teacher* in the task of forgetting 500 images, with a $\bar{\Delta}$ improvement from $8.28\%$ to $5.52\%$. Similarly, in the task of forgetting 4000 images, we achieve an $\bar{\Delta}$ boost from $15.94\%$ to $8.85\%$.

### A.3 COMPARISON WITH THE *Fisher Forgetting* METHOD

In addition to the four methods outlined in the main body of our manuscript, we also compare our method with the *Fisher Forgetting* method Golatkar et al. (2020b). We evaluate to forget 100 images from each class in the CIFAR-10 dataset. As observed from Table.13, the Fisher Forgetting significantly undermines the performance of the model, while our method achieves effective forgetting while maintaining the model's performance.

Table 8: Comparison with the **ViT** model on **CIFAR-10** dataset. Regarding $M_o$, its average accuracy over 10 classes is $84.45\%$.

| Removal Class | Gold model $ACC^g_{D_{test}}$ | $ACC^g_{D_f}$ | Fine-tuning $ACC_{D_{test}}$ | $ACC_{D_f}$ | Negative Gradient $ACC_{D_{test}}$ | $ACC_{D_f};\Delta$ | Random Labeling $ACC_{D_{test}}$ | $ACC_{D_f}$ | Bad Teacher $ACC_{D_{test}}$ | $ACC_{D_f};\Delta$ | Ours $ACC_{D_{test}}$ | $ACC_{D_f};\Delta$ |
|---|---|---|---|---|---|---|---|---|---|---|---|---|
| Class 0 | 84.01 | 92 | 84.70 | 100 | 84.46 | 62; 30 | 84.16 | 4 | 83.49 | 92; 0 | 82.94 | 88; 4 |
| Class 1 | 84.26 | 96 | 84.71 | 100 | 84.58 | 76; 20 | 84.23 | 9 | 83.47 | 87; 9 | 83.37 | 96; 0 |
| Class 2 | 85.66 | 88 | 84.54 | 100 | 84.58 | 60; 28 | 84.33 | 1 | 82.90 | 82; 6 | 83.97 | 91; 3 |
| Class 3 | 84.80 | 70 | 84.67 | 100 | 84.35 | 41; 29 | 84.17 | 0 | 83.02 | 87; 17 | 84.16 | 74; 4 |
| Class 4 | 85.24 | 87 | 84.41 | 100 | 84.56 | 51; 36 | 84.30 | 0 | 83.19 | 87; 0 | 83.85 | 86; 1 |
| Class 5 | 84.32 | 78 | 84.39 | 100 | 84.37 | 54; 24 | 84.13 | 0 | 83.01 | 86; 8 | 83.67 | 80; 2 |
| Class 6 | 83.39 | 85 | 84.78 | 100 | 84.54 | 71; 14 | 84.14 | 3 | 83.38 | 93; 8 | 83.85 | 90; 5 |
| Class 7 | 84.97 | 85 | 84.74 | 100 | 84.63 | 66; 19 | 84.29 | 2 | 82.51 | 85; 0 | 83.85 | 79; 6 |
| Class 8 | 85.13 | 92 | 85.7 | 100 | 84.54 | 72; 20 | 84.28 | 3 | 83.37 | 90; 2 | 83.65 | 89; 3 |
| Class 9 | 84.38 | 93 | 85.65 | 100 | 84.50 | 74; 19 | 84.08 | 2 | 83.41 | 90; 3 | 83.62 | 88; 5 |
| Avg | 84.61 | 86.6 | 84.83 | 100 | 84.51 | 62.7; **23.9** | 84.21 | 2.4 | 83.05 | 87.9; **5.3** | 84.07 | 86.1; **3.3** |

Table 9: Comparison with the **ViT** model on **CIFAR-100** dataset. We randomly sample 17 classes for evaluation. The average accuracy of $M_o$ over 100 classes is $83.04\%$.

| Removal Class | Gold model $ACC^g_{D_{test}}$ | $ACC^g_{D_f}$ | Fine-tuning $ACC_{D_{test}}$ | $ACC_{D_f}$ | Negative Gradient $ACC_{D_{test}}$ | $ACC_{D_f};\Delta$ | Random Labeling $ACC_{D_{test}}$ | $ACC_{D_f}$ | Bad Teacher $ACC_{D_{test}}$ | $ACC_{D_f};\Delta$ | Ours $ACC_{D_{test}}$ | $ACC_{D_f};\Delta$ |
|---|---|---|---|---|---|---|---|---|---|---|---|---|
| road | 81.97 | 95 | 83.43 | 100 | 80.51 | 41; 54 | 83.04 | 0 | 81.80 | 95; 0 | 82.13 | 100; 5 |
| turtle | 81.88 | 80 | 83.45 | 100 | 80.54 | 22; 58 | 80.54 | 0 | 82.00 | 65; 15 | 80.61 | 83; 3 |
| chimpanzee | 82.26 | 96 | 83.43 | 100 | 80.43 | 28; 68 | 83.52 | 0 | 81.30 | 81; 15 | 82.42 | 100; 4 |
| orchid | 81.70 | 93 | 83.42 | 100 | 80.37 | 25; 68 | 83.08 | 0 | 80.59 | 72; 21 | 81.96 | 87; 6 |
| rabbit | 82.11 | 75 | 83.46 | 100 | 80.44 | 14; 63 | 82.95 | 0 | 81.09 | 48; 27 | 82.42 | 89; 14 |
| forest | 82.49 | 77 | 83.45 | 100 | 80.37 | 24; 54 | 83.20 | 0 | 80.93 | 76; 1 | 82.20 | 90; 13 |
| possum | 82.39 | 78 | 83.41 | 100 | 80.73 | 16; 62 | 83.36 | 0 | 80.98 | 57; 21 | 82.38 | 84; 6 |
| fox | 81.67 | 90 | 83.43 | 100 | 80.43 | 24; 66 | 83.11 | 0 | 80.55 | 92; 2 | 82.19 | 86; 4 |
| house | 82.26 | 86 | 83.42 | 100 | 80.59 | 24; 59 | 83.35 | 0 | 81.03 | 83; 3 | 82.31 | 67; 19 |
| mushroom | 81.22 | 88 | 83.45 | 100 | 80.73 | 20; 66 | 83.27 | 0 | 81.43 | 94; 6 | 82.43 | 95; 7 |
| chair | 83.51 | 88 | 83.43 | 100 | 80.67 | 39; 49 | 83.11 | 0 | 81.45 | 95; 7 | 82.41 | 94; 6 |
| tiger | 82.81 | 87 | 83.47 | 100 | 80.54 | 35; 52 | 83.15 | 0 | 81.11 | 94; 7 | 81.57 | 84; 3 |
| snail | 83.79 | 90 | 83.41 | 100 | 80.54 | 18; 72 | 82.19 | 0 | 80.80 | 86; 4 | 82.4 | 94; 4 |
| worm | 83.22 | 92 | 83.41 | 100 | 80.61 | 38; 54 | 83.04 | 0 | 81.22 | 96; 4 | 82.56 | 99; 7 |
| beetle | 83.12 | 80 | 83.39 | 100 | 80.01 | 15; 65 | 83.15 | 0 | 81.32 | 92; 12 | 81.6 | 97; 17 |
| beaver | 83.05 | 67 | 83.44 | 100 | 80.37 | 6; 61 | 83.50 | 0 | 81.37 | 71; 4 | 81.76 | 71; 4 |
| bed | 83.30 | 81 | 83.44 | 100 | 80.57 | 25; 56 | 83.25 | 0 | 81.29 | 88; 7 | 81.7 | 89; 8 |
| avg | 82.51 | 84.71 | 83.43 | 100 | 80.50 | 39.2; **45.51** | 83.10 | 0 | 81.19 | 81.47; **8,64** | 82.06 | 88.76; **6.23** |

| Gold model $ACC^g_{D_{test}}$ | $ACC^g_{D_f}$ | Fine-tuning $ACC_{D_{test}}$ | $ACC_{D_f}$ | Negative Gradient $ACC_{D_{test}}$ | $ACC_{D_f};\Delta$ | Random Labeling $ACC_{D_{test}}$ | $ACC_{D_f}$ | Bad Teacher $ACC_{D_{test}}$ | $ACC_{D_f};\Delta$ | Ours $ACC_{D_{test}}$ | $ACC_{D_f};\Delta$ |
|---|---|---|---|---|---|---|---|---|---|---|---|
| 85.81 | 94 | 87.09 | 100 | 85.91 | 63; 31 | 84.26 | 0 | 84.51 | 91; 3 | 85.86 | 90; 4 |
| 85.31 | 83 | 86.89 | 100 | 85.72 | 67; 14 | 85.52 | 0 | 83.31 | 72; 9 | 86.49 | 82; 1 |
| 84.61 | 91 | 86.60 | 100 | 85.21 | 60; 38 | 85.79 | 1 | 84.21 | 80; 11 | 86.16 | 92; 1 |
| 85.31 | 87 | 86.50 | 100 | 85.11 | 58; 39 | 84.50 | 0 | 84.61 | 89; 2 | 86.59 | 85; 2 |
| 85.11 | 97 | 86.70 | 100 | 85.50 | 66; 30 | 85.89 | 0 | 85.91 | 94; 4 | 85.50 | 94; 3 |
| 85.02 | 98 | 86.79 | 100 | 85.01 | 56; 40 | 85.95 | 0 | 85.21 | 85; 13 | 86.15 | 91; 7 |
| 85.71 | 95 | 86.70 | 100 | 84.64 | 53; 31 | 84.84 | 0 | 85.81 | 86; 9 | 86.67 | 94; 1 |
| 85.41 | 81 | 86.79 | 100 | 85.38 | 63; 16 | 84.61 | 0 | 84.61 | 68; 13 | 87.28 | 90; 9 |
| 85.42 | 87 | 86.89 | 100 | 84.93 | 48; 39 | 85.15 | 0 | 83.21 | 55; 33 | 87.08 | 87; 0 |
| 84.82 | 94 | 86.60 | 100 | 85.16 | 52; 42 | 83.85 | 0 | 83.61 | 84; 10 | 86.38 | 96; 2 |
| 85.25 | 90.7 | 86.76 | 100 | 85.23 | 60.5; **30.2** | 85.01 | 0.1 | 84.50 | 80.4; **10.7** | 86.41 | 90.1; **3** |

Table 10: Comparison on **VGGFaces2** dataset. Regarding $M_o$, its average accuracy over 10 classes is $87.01\%$.

Table 11: Comparison on CIFAR-10 dataset with the size of $D_f$ as **500**. Regarding $M_o$, its average accuracy over 10 classes is $85.37$.

| Gold model $ACC^g_{D_{test}}$ | $ACC^g_{D_f}$ | Fine-tuning $ACC_{D_{test}}$ | $ACC_{D_f}$ | Negative Gradient $ACC_{D_{test}}$ | $ACC_{D_f};\Delta$ | Random Labeling $ACC_{D_{test}}$ | $ACC_{D_f}$ | Bad Teacher $ACC_{D_{test}}$ | $ACC_{D_f};\Delta$ | Ours $ACC_{D_{test}}$ | $ACC_{D_f};\Delta$ |
|---|---|---|---|---|---|---|---|---|---|---|---|
| 85.77 | 87.6 | 85.08 | 100 | 84.90 | 40.4; 47.2 | 84.44 | 0 | 83.51 | 97.0; 9.4 | 83.60 | 90.2; 2.6 |
| 85.56 | 94.6 | 85.02 | 100 | 84.93 | 44.2; 50.4 | 84.76 | 0.2 | 84.25 | 100; 5.2 | 84.58 | 98.2; 3.6 |
| 85.96 | 80.4 | 85.07 | 100 | 84.66 | 34.0; 46.4 | 84.04 | 0 | 83.68 | 96.6; 16.2 | 84.35 | 90.5; 10.1 |
| 85.89 | 68.6 | 84.93 | 100 | 84.89 | 29.2; 39.4 | 85.45 | 0 | 79.75 | 74.8; 9.3 | 85.37 | 79.4; 10.6 |
| 84.93 | 81.4 | 84.91 | 100 | 84.75 | 30.8; 50.4 | 84.89 | 0 | 83.45 | 97.8; 16.4 | 84.43 | 86.2; 4.8 |
| 85.67 | 74.5 | 85.03 | 100 | 84.94 | 32.8; 41.7 | 85.08 | 0 | 82.90 | 90.8; 16.3 | 86.12 | 78.3; 3.8 |
| 85.82 | 87.6 | 85.05 | 100 | 80.46 | 10.0; 77.6 | 85.07 | 0 | 82.01 | 93.4; 5.8 | 85.47 | 89.2; 1.6 |
| 85.55 | 91.2 | 84.92 | 100 | 84.72 | 33.8; 57.4 | 84.30 | 0.2 | 78.22 | 85.2; 6.0 | 84.40 | 98.2; 7.0 |
| 85.44 | 94.8 | 84.93 | 100 | 85.05 | 44.4; 50.5 | 82.36 | 1.4 | 80.81 | 97.8; 3.0 | 84.72 | 97.0; 2.2 |
| 85.83 | 93.0 | 84.97 | 100 | 85.02 | 38.0; 55.0 | 85.11 | 0 | 84.19 | 99.6; 6.6 | 84.93 | 94.6; 1.6 |
| 85.64 | 85.39 | 84.99 | 100 | 84.32 | 33.76; **51.60** | 83.92 | 0.18 | 82.27 | 93.28; **8.28** | 84.81 | 90.18; **5.52** |

Table 12: Comparison on CIFAR-10 dataset with the size of $D_f$ as **4000**. Regarding $M_o$, its average accuracy over 10 classes is 85.37.

| Removal Class | Gold model | | Finetune | | Negtive Gradient | | Random Labeling | | Bad Teacher | | Ours | |
| | $\text{ACC}^g_{D_{test}}$ | $\text{ACC}^g_{D_f}$ | $\text{ACC}_{D_{test}}$ | $\text{ACC}_{D_f}$ | $\text{ACC}_{D_{test}}$ | $\text{ACC}_{D_f};\Delta$ | $\text{ACC}_{D_{test}}$ | $\text{ACC}_{D_f}$ | $\text{ACC}_{D_{test}}$ | $\text{ACC}_{D_f};\Delta$ | $\text{ACC}_{D_{test}}$ | $\text{ACC}_{D_f};\Delta$ |
|---|---|---|---|---|---|---|---|---|---|---|---|---|
| class 0 | 84.01 | 62.44 | 85.31 | 100 | 78.90 | 22.42; 40.02 | 76.12 | 1.83 | 79.42 | 60.67; 1.77 | 82.95 | 61.65; 0.79 |
| class 1 | 84.26 | 80.25 | 85.12 | 100 | 76.61 | 22.97; 57.28 | 76.44 | 0.83 | 78.54 | 51.55; 28.70 | 83.61 | 85.92; 5.67 |
| class 2 | 82.40 | 42.65 | 85.01 | 100 | 79.32 | 13.40; 39.25 | 76.99 | 0.38 | 82.42 | 77.45; 34.80 | 80.34 | 22.53; 20.12 |
| class 3 | 82.56 | 24.80 | 85.36 | 100 | 80.78 | 8.97; 15.83 | 77.65 | 0.05 | 78.22 | 28.70; 3.90 | 79.02 | 12.42; 12.38 |
| class 4 | 82.20 | 48.05 | 85.26 | 100 | 79.13 | 12.05; 36.00 | 76.34 | 0.03 | 79.91 | 46.37; 1.68 | 79.22 | 26.70; 21.35 |
| class 5 | 82.99 | 38.52 | 85.34 | 100 | 80.34 | 11.55; 26.97 | 77.11 | 0.03 | 80.92 | 70.47; 31.95 | 82.23 | 29.30; 9.22 |
| class 6 | 84.04 | 65.90 | 85.30 | 100 | 77.42 | 11.37; 54.53 | 76.53 | 0.10 | 81.79 | 37.80; 28.10 | 81.07 | 59.87; 6.03 |
| class 7 | 83.38 | 62.87 | 85.23 | 100 | 77.99 | 20.17; 42.70 | 76.19 | 0.03 | 77.02 | 72.92; 10.05 | 82.51 | 62.82; 0.05 |
| class 8 | 84.18 | 77.65 | 85.25 | 100 | 77.90 | 30.17; 47.48 | 75.92 | 0.80 | 80.82 | 83.85; 6.20 | 83.11 | 72.42; 5.23 |
| class 9 | 84.38 | 73.55 | 85.29 | 100 | 76.00 | 18.52; 55.03 | 75.13 | 0.13 | 80.06 | 61.32; 12.23 | 81.99 | 81.62; 7.70 |
| Avg | 83.44 | 57.38 | 84.79 | 100 | 78.41 | 17.15; **43.89** | 76.44 | 0.42 | 79.91 | 63.65; **15.94** | 81.60 | 51.32; **8.85** |

Table 13: The *Fisher Forgetting* method can potentially result in catastrophic forgetting, severely compromising the performance of the model.

| | Gold model | | Fisher Forgetting | | Ours | |
| model | $\text{ACC}^g_{D_{test}}$ | $\text{ACC}^g_{D_f}$ | $\text{ACC}_{D_{test}}$ | $\text{ACC}_{D_f}$ | $\text{ACC}_{D_{test}}$ | $\text{ACC}_{D_f}$ |
|---|---|---|---|---|---|---|
| Resnet18 | 85.61 | 92 | 10.37 | 0 | 84.60 | 96 |
| AllCNN | 87.00 | 94 | 10.07 | 2 | 87.03 | 89 |

