# OpenReview forum: "Towards Aligned Data Forgetting via Twin Machine Unlearning"
_ICLR.cc/2025/Conference — Submitted to ICLR 2025_

### Official Review · Reviewer_zYUF · 2024-10-25

**Soundness:** 2
**Presentation:** 2
**Contribution:** 2
**Rating:** 3
**Confidence:** 5

**Summary:**

The paper proposes an aligned data forgetting method via twin machine unlearning. The research problem is interesting and hot. However, the writing of the paper needs to be heavily polished, and the logic also needs to be heavily proofread. For the reviewer, the solution is not sound from two aspects: (1) The authors assume the gold model M_o and the twin model M_t are known; however, knowing these two models consumes the exact cost of original training and retraining, both needing to train two models. (2) Unlearning D_f^h but optimizing D_f^e is not reasonable, because both datasets belong to D_f, which needs unlearning.

**Strengths:**

Strengths:
1. The research problem is interesting and hot.
2. The experiments looks sufficient.

**Weaknesses:**

Weaknesses:
1. There are lots of writing and logic confusion. For example, in Figure 1, the left half is "forget D_f", the right half is "forget D_{test}". It is weird why we don't forget D_f in both problems to keep the logic aligned. It looks like the authors created a new problem to make the solution reasonable. However, from line 069 to 084, the paper seems aiming to forget $D_f$.

2. For the second contribution, what is the aim to distinguish easy and hard samples, D_f^e and D_f^h, as the new unlearning problem is to unlearning D_{test}.

3. The optimization in Equation (2) and (3) seems confused. Both D_f^h and D_f^e are in D_f, which needs to be forgotten. Why do we maximize the loss of D_f^h but continue to optimize the loss of D_f^e? Continuing to optimize the loss of D_f^e keeps learning of D_f^e rather than unlearning it.

4. The twin unlearning problem is to unlearn D_{test}, however, it is not clear how to unlearn the D_{test} in the paper and how M_t is achieved. Equation (1) seems to assume the M_o is already achieved. However, we should know that for the twin unlearning problem, the cost to achieve M_o is similar to retraining the model for the original unlearning problem.

**Questions:**

No additional questions

---

### Official Review · Reviewer_xA6C · 2024-11-04

**Soundness:** 2
**Presentation:** 2
**Contribution:** 2
**Rating:** 5
**Confidence:** 4

**Summary:**

This paper presents an interesting approach called "Twin Machine Unlearning (TMU)" to enhance the "alignment" between the performance of the unlearned model and the retrained model (referred to as the gold model in this paper). The core idea of TMU is that a well-aligned unlearning algorithm should specifically decrease the classification accuracy on part of the samples in the forget set, rather than all samples. To achieve this, the authors propose dividing the forget set into two subsets: "easy" samples, which the retrained model can classify correctly, and "hard" samples, which it cannot. The main technical challenge is training a binary classifier to effectively distinguish between easy and hard samples, given that the retrained model is typically unknown. The TMU solution addresses this by first training a "twin model" $M_t$ using an additional dataset $D_{test}$, then extracting three discriminative features with M_t to predict easy/hard labels on the forget set $D_f$.

**Strengths:**

S1. The paper is well-motivated, addressing the problem of enhancing the "alignment" between the performance of the unlearned model and the retrained model.

S2. The introduction of the novel concept "Twin Machine Unlearning" is interesting and appealing.

S3. Overall, the authors clearly articulate the challenges of their research and the corresponding solutions, although some detailed explanations are missing and require further clarification (see Weaknesses and Questions).

**Weaknesses:**

W1. The statement that "In prior unlearning methods, the concept of ‘data forgetting’ is often interpreted and implemented as achieving zero classification accuracy on such data" is somewhat partial and misleading. In fact, only class-level forgetting may require achieving zero classification accuracy on the forget set. For general random data forgetting task, most prior unlearning studies also consider the retrained model (i.e., the gold model in this paper) as the baseline and aim for optimal alignment in terms of model performance.

W2. Some detailed explanations are missing and need further clarifications (see Questions).

W3. Typos:
1. Line 318: “We have build a specific labeled dataset”: build -> built.
2. Line 231: “Eq.equation 3”
3. “Negtive” -> “Negative” in Table 1, Table 2, and Table 12.
4. Consider adding a dot in f(·) rather than just using f().

**Questions:**

Q1. Given the inherent randomness in predictions for samples near the decision boundary, it would be beneficial for the authors to clarify why this binary (easy-vs-hard) label can accurately reflect the model's generalization ability with respect to a target sample. Intuitively, additional restrictions on the predicted outputs should be considered when distinguishing between easy and hard samples. For example, the confidence score on the ground-truth label should be sufficiently high to indicate that the sample is "indeed" easy to classify.

Q2. Please clarify if I have missed something. It seems there is no explanation on how to obtain the external dataset $D_{test}$ in this paper. The authors only state that "$D_{test}$ can be seen as analogous to Mt’s forgetting data" and "It is usually assumed that $D_{test}$ and $D_f$ have independent and identical distribution (I.I.D)". These statements are confusing. How can we know the distribution of the forgotten data in advance? The authors are strongly encouraged to provide more clarification on constructing $D_{test}$.

Q3. In line 321 “We concatenate the three discriminative features as the input of the binary classifier”, which set of samples are you referring to here, the forget set $D_f$ or the test set $D_{test}$? Please clarify this point.

Q4. The statement "our approach can afford the examination of data forgetting from MIA" in line 462 seems somewhat overstated. From Table 1, aside from the Negative Gradient and Random Label approaches, which consistently achieve a lower MIA attack success rate (ASR) among all methods, the ASR of "Ours" does not present a significantly lower value compared to other baselines.

Q5: In line 463-464: “It is worthy to note that the Negative Gradient achieve better performance than our approach in terms of MIA assessment. This is because the Bad Teacher inherently aims to disrupt accuracy on Df, ….”, is there a causal relationship between these two sentences?

---

> ### Author Response · Authors · 2024-11-27
> **Responses to questions**
>
> Q2: Yes, we will clarify this further. In fact, we implicitly adopt a conventional machine learning framework, where the entire dataset is split into training and testing sets based on a predefined ratio. The $D_{test}$ mentioned in our paper refers to this "testing set." Within this conventional framework, it is assumed that the training and testing sets are independently and identically distributed (I.I.D).
>
> Q4: We need to consider all metrics simultaneously. The Negative Gradient and Random Label approaches perform poorly on $ACC_{Df}$, making the Bad Teacher approach the real competitor.

---

### Official Review · Reviewer_qfUX · 2024-11-07

**Soundness:** 2
**Presentation:** 2
**Contribution:** 2
**Rating:** 5
**Confidence:** 4

**Summary:**

This paper introduces the Twin Machine Unlearning (TMU) approach to induce unlearning while retaining the model's generalization ability *without* the need to re-train it from scratch. Additionally, the authors introduce a noise-perturbed fine-tuning algorithm to enhance the unlearned model's ability to mitigate membership inference attacks (MIAs), which could determine whether the unlearned data was once a part of the target model's training.

**Strengths:**

* The authors devise a new approach to machine unlearning conscious of the computational burden of re-training the target model from scratch.

* Using multiple discriminative features to compute the generalisation label for each sample in $D_{f}$ and $D_{test}$. From Table 3, it is clear that no one metric dominates across all the classes. Using multiple features helps bolster the accuracy of the binary classifier used to label the sample as easy or hard to fit.

* The method is straightforward to implement. A systematic evaluation against multiple prior approaches demonstrates that it surpasses the latter in terms of the alignment achieved between the unlearned model and the gold model on the forgotten data.

* The concept of using noise to mitigate MIAs has been extensively explored in the field of differential privacy but the authors have developed a novel method to actively incorporate noise in their algorithm with the explicit aim to make the unlearned model resistant to MIAs.

**Weaknesses:**

* TMU heavily relies on $D_{test}$ and $D_f$ being identically distributed. The authors do not elaborate on how the algorithm's efficacy may vary if this assumption does not hold. For example, if the two data sets differ in terms of the distribution of targeted classes, it is unclear how much it will affect the performance of TMU.

* The presentation of the Results section could have been better. For some of the tables and figures, e.g. Table 4 and Figure 3 in the main text, it is unclear which experiments were used to compute the results.

* Despite it being one of the key contributions of the paper, the authors do not provide much detail in Results (4.2) or Ablation Study (4.3), beyond Table 4, on the effect of the unlearned model's vulnerability to MIAs with the noise-perturbed fine-tuning scheme.

* The suggested approach offers weak resistance to MIA on $D_f^e$ as observed by the high MIA ASR in the results (with or without noise). The authors could have attempted to probe this further.

* Enhanced population-based MIA [2] uses a fixed threshold across all samples. It will likely be confident about its predictions for non-members but not so much for the members. As such it is a subpar approach to MIA compared to one where the threshold is sample-dependent, e.g. reference model-based MIAs such as LiRA [1].

* Tarun et al. [3] suggested *relearn time* as a metric for evaluating whether a model still holds any information corresponding to the forgotten data. In the case of the twin unlearning problem, the gold model is known. You could have used it as an additional metric to evaluate the efficacy of TMU.

**[1]** N. Carlini, S. Chien, M. Nasr, S. Song, A. Terzis and F. Tramèr, Membership Inference Attacks From First Principles, IEEE S&P 2022. doi: 10.1109/SP46214.2022.9833649.

**[2]** Jiayuan Ye, Aadyaa Maddi, Sasi Kumar Murakonda, Vincent Bindschaedler, and Reza Shokri, Enhanced Membership Inference Attacks against Machine Learning Models, ACC CCS'22). https://doi.org/10.1145/3548606.3560675.

**[3]** A. K. Tarun, V. S. Chundawat, M. Mandal and M. Kankanhalli, Fast Yet Effective Machine Unlearning, IEEE Neural Networks and Learning Systems. doi: 10.1109/TNNLS.2023.3266233.

**Questions:**

*Questions:*

1. What was the author's rationale behind choosing a subpar attack for membership inference?

2. Was there a reason why *relearn time* was not used as an evaluation metric in your study?

*Suggestions:*

* Concerning the attack success rate (ASR), Carlini et al [1] proposed computing the ASR at a low false positive rate (FPR) to evaluate the efficacy of MIAs granularly. Average ASR is computed over all FPRS and hence, it could be a misleading metric. With ASR at a low FPR, an attack is considered strong if an attacker can correctly identify a large number of training samples at a very low FPR. I would recommend using MIA ASR at low FPRs as a metric for assessing the vulnerability of the unlearned model against MIA on $D_f^e$.

* Please give the paper a thorough re-read for a spellcheck. I identified several typos while reading the paper, e.g. on Page 8, in the tables, Negative has been spelled as "Negtive".

* On Page 9, I believe what you're trying to say is that since the Negative Gradient approach works by deliberately undermining the model's performance on $D_f$, it is more resilient against MIA than your approach. You seem to have swapped Negative Gradient with Bad Teacher midway through the paragraph. Do give the paper another read to correct for such mistakes.

**[1]** N. Carlini, S. Chien, M. Nasr, S. Song, A. Terzis and F. Tramèr, Membership Inference Attacks From First Principles, IEEE S&P 2022. doi: 10.1109/SP46214.2022.9833649.

**NOTE**: *If the authors could address the highlighted concerns (weaknesses/ questions), I am open to reconsidering my initial Rating for the paper.*

---

> ### Author Response · Authors · 2024-11-26
> **Helpful comments**
>
> We appreciate the reviewer’s comments. We are grateful for the suggestion to use TPR@low FPR as a metric. We will adopt this metric to evaluate our approach.

---

> > ### Comment · Reviewer_qfUX · 2024-11-26
> > **Revising the manuscript**
> >
> > Dear Authors,
> > Do you intend to revise the manuscript to include TPR@low FPR as a metric for the MIA experiments?

---

> > > ### Author Response · Authors · 2024-11-26
> > > **Sure**
> > >
> > > Yes, we have carefully read the paper [1] and agree that this metric provides a more accurate reflection of MIA.
> > >
> > > By the way, I believe Ye's work [2] has addressed the FPR issue to some extent (as mentioned in [1], "The attack in the concurrent work of Ye et al. is close in spirit to ours"). Therefore, utilizing Ye's work (instead of Carlini's work) as an MIA may not be ideal, but it can still serve to effectively validate the efficacy of our approach.

---

> > > > ### Comment · Reviewer_qfUX · 2024-11-26
> > > > **Seeking clarity over the nature of MIA used**
> > > >
> > > > Ye et al.[2] discusses multiple versions of MIA. Attack-P in the paper is a shadow model-free approach whereas attack-S, -R and -D use shadow models. Out of these, Attack-R is most similar to LiRA. Given that you do not mention training shadow models in your manuscript, I am unclear as to which one of these attacks was used in your paper. Could you clarify this for me?

---

> > > > > ### Author Response · Authors · 2024-11-26
> > > > > **The strongest variant**
> > > > >
> > > > > We leverage Attack-D in our approach, since it is the strongest one among all variants. Attack-D possesses access to all prior knowledge, i.e., Record and Model, as shown in Table.1 in [2].

---

> > > > > > ### Comment · Reviewer_qfUX · 2024-11-26
> > > > > > **Thanks for the clarification**
> > > > > >
> > > > > > I appreciate you making that clear to me. Kindly modify the manuscript to include this information. It is not easy to assume that you are using the Attack-D given that you make no mention of it.

---

> > > > > > > ### Author Response · Authors · 2024-11-26
> > > > > > > **Thank you**
> > > > > > >
> > > > > > > Yes, we appreciate your comments indeed :-)

---

### Official Review · Reviewer_FeJE · 2024-11-09

**Soundness:** 2
**Presentation:** 2
**Contribution:** 2
**Rating:** 5
**Confidence:** 3

**Summary:**

This paper addresses the alignment problem in machine unlearning, ensuring that the unlearned model (the "gold model") still predicts correct labels for the unlearning subsets, thanks to the generalization abilities of machine learning models. To solve this, the authors propose a twin unlearning strategy, where a twin model is fine-tuned from the original model using a hold-out set. They then train a binary classifier on the forgetting data from this process, which partitions the data into a "hard" subset, where forgetting is difficult, and an "easy" subset, where forgetting is straightforward. The original model is fine-tuned to reduce classification accuracy on the hard subset while maintaining accuracy on the easy subset. This approach improves alignment in data forgetting and provides better resilience to Membership Inference Attacks (MIA) compared to previous methods.

**Strengths:**

- The alignment issue in machine unlearning is well-identified and motivated.
- The twin model strategy seems feasible in practice.

**Weaknesses:**

- The presentation needs to be improved. The authors should provide more detailed explanations and justifications for their choices. For instance, how to set the boundary between hard/easy labels in practice? Why a well-aligned unlearning algorithm should decrease the accuracy on hard samples?
- The evaluation is not convincing. The authors should provide more comprehensive experiments and study the key factors that affect the alignment.
- The limitations of the proposed method are not well-discussed. For example, the noise-perturbed fine-tuning scheme may not be optimal in all settings, and can be challenging to tune in practice. The proposed method may not be scalable to large datasets or complex models due to the computational overhead of training the twin model and binary classifier. Additionally, the proposed approach assumes that the forgetting data is uniformly distributed over the feature space, which may not always be the case.

**Questions:**

1. Why a well-aligned unlearning algorithm should decrease the accuracy on hard samples?
2. How does the twin model strategy help in achieving aligned machine unlearning?
3. What are the key factors that affect the alignment performance of the twin model?
4. How many times did you repeat the experiments to ensure the robustness of the proposed method?
5. What are the limitations of the proposed twin unlearning strategy, and how do they impact the overall effectiveness of the unlearning process?
6. Are there any plans or how easy to extend this work to different types of machine learning models (like language models)?
7. How sensitive is the proposed method to the choices of hold-out sets that are used to train the twin model?

---

> ### Author Response · Authors · 2024-11-27
> **extending this work to LLM**
>
> Thank you for your valuable suggestions. It is intriguing to contemplate extending this work to LLMs, and we intend to explore this direction in the future. However, such an extension would necessitate a fundamentally different approach, as LLMs are generative models, whereas our current work focuses on discriminative models.

---

### Meta-Review · Area_Chair_FgoJ · 2024-12-16

**Metareview:**

The paper proposes a novel unlearning method that empirically considers the difference to a model trained without the data meant to be unlearned.

The proposed idea appears novel and potentially interesting.

However, the reviewers criticise the work for weak evaluation of the proposed method.

The authors have promised to address the reviewers' comments, but as no revision appears to be submitted, the paper is not ready for publication at ICLR.

**Additional Comments On Reviewer Discussion:**

As far as I can tell, the authors did not make any changes to the manuscript (at least none were marked) so all revisions were just promises.

---

### Decision · Program_Chairs · 2025-01-22

Reject